# A Redo Percutaneous Emergency Intervention of Left Ventricular Assist Device Graft Occlusion

**DOI:** 10.3390/ijerph19105976

**Published:** 2022-05-14

**Authors:** Rocco Edoardo Stio, Marina Comisso, Luca Paolucci, Silvio Coletta, Vincenzo Cesario, Michele Gioia, Marco Stefano Nazzaro, Guglielmo Saitto, Carlo Contento, Emilio D’Avino, Francesco De Felice, Domenico Gabrielli, Francesco Musumeci

**Affiliations:** 1Interventional Cardiology Unit, Department of Heart and Vessels, Division of Interventional Cardiology, San Camillo-Forlanini Hospital, 00152 Rome, Italy; l.paolucci@unicampus.it (L.P.); silcoletta@gmail.com (S.C.); vicesario91@gmail.com (V.C.); michelegioia10@gmail.com (M.G.); msnazzaro@gmail.com (M.S.N.); f.defelice1966@gmail.com (F.D.F.); dgabrielli@scamilloforlanini.rm.it (D.G.); 2Department of Heart and Vessels, Cardiac Surgery and Heart Transplantation Center, San Camillo-Forlanini Hospital, 00152 Rome, Italy; marina.comisso@gmail.com (M.C.); guglielmo.saitto@gmail.com (G.S.); fr.musumeci@gmail.com (F.M.); 3Department of Cardiovascular Anaesthesia, San Camillo-Forlanini Hospital, 00152 Rome, Italy; carlo959@gmail.com (C.C.); edavino@scamilloforlanini.rm.it (E.D.)

**Keywords:** left ventricular assist device, percutaneous coronary intervention, heart failure

## Abstract

In patients with advanced heart failure (HF), left ventricular assist devices (LVADs) have demonstrated to be effective in improving the quality of life and reducing further hospitalizations. Although uncommon, LVAD outflow graft obstruction (OGO) is a potentially life-threatening complication and percutaneous treatment has been proposed as a standard intervention strategy in such cases. We report the case of a 69 year old man admitted due to LVAD failure causing unstable HF. Past medical history included percutaneous intervention on the outflow graft with stent implantation one year before. The patient was under chronic treatment with vitamin K antagonists (VKA). Emergent percutaneous angiography was performed, showing recurrent OGO due to thrombosis located at a kinking site, distally to the previously treated segment. Using distal anchoring technique, a balloon-expandable 10 × 79 mm endoprosthesis (GORE^®^ Viabahn^®^ VBX) was effectively positioned and post-dilated. Final angiography confirmed the patency of the stent implanted one-year before. Despite the procedure succeeding in restoring LVAD function, the patient died due to septic shock ten days after. Our case suggests that recurrent OGO can be effectively treated with percutaneous redo and that long-term stent patency can be achieved with a standard antithrombotic treatment, despite further thrombotic events in other segments of the graft are still possible (especially at the kinking site). Moreover, other noncardiac conditions as infective complications, can dramatically impact the clinical course and lead to unfavorable outcomes.

## 1. Introduction

Despite the major advances in the medical and interventional treatment of heart failure (HF) [1], end-stage HF is still a leading cause of morbidity and mortality [2]. In these patients, left ventricular assist devices (LVADs), as a bridge to destination or bridge to transplantation therapy, have demonstrated to be effective in improving the quality of life and reducing further hospitalizations [3]. Nevertheless, patients with impaired ejection fraction treated with LVAD can often experience recurrent HF and, in such cases, device malfunctioning should always be suspected, especially when HF occurs late after implantation (>4 weeks) [4]. Despite being considered an uncommon event [3], outflow graft obstruction (OGO) is progressively recognized as a relevant cause of LVAD failure and its incidence may be superior to 5% [5]. In these patients, percutaneous treatment with stent implantation can be a life-saving intervention [6] and it has been proposed as a first step strategy in order to avoid high risk surgical redo [7]. Nevertheless, data regarding long term clinical outcome and stent patency in patients suffering OGO previously treated with a percutaneous approach are critically lacking. In this report, we describe the case of a patient who underwent a percutaneous redo due to recurrent OGO.

## 2. Case Report

A 69 year old male with advanced HF (LV ejection fraction: 25%) due to severe familial dilated cardiomyopathy was admitted to our emergency department, complaining of severe dyspnea and asthenia. The symptoms had started one week previously and had progressively worsened. Three years before, the patient had undergone surgical aortic valve replacement for severe aortic regurgitation (Inspiris Resilia TM 26 mm, Edwards Lifesciences LLC, Irvine, CA, USA) combined with a tricuspid valve ring annuloplasty (Contour 3D TM, Medtronic, Dublin, Ireland) and followed by LVAD implantation (HeartMate III TM, Abbott, NC, USA) as a destination therapy. One year after this, the patient experienced LVAD failure caused by a proximal thrombosis of the outflow graft, which was effectively treated with a percutaneous intervention using a 10 × 59 mm balloon-expandable covered stent (ADVANTA V12-8 Fr compatible covered stent, Getinge AB) [8]. In Figure 1A,B pre- and postprocedural angiographies of the first procedure are shown.

Following this first LVAD intervention, a dual antithrombotic regimen with aspirin and a vitamin K antagonist (VKA) was administered. However, antiplatelet therapy was suspended few months after due to a major neurological bleeding. At admission, the patient showed unstable clinical conditions (systemic blood pressure of 90/60 mmHg, a heart rate of 130 beats per minute, a respiratory rate of 20 breaths per minute, arterial oxygen saturation of 85%). Laboratory markers showed elevated serum lactate and impaired renal function, without signs of ongoing hemolysis. Due to the impeding cardiogenic shock, the patient was admitted to intensive care unit (ICU). Echocardiographic evaluation confirmed the presence of severe LV dysfunction (ejection fraction < 25%) and dilation. Moreover, severe right ventricular (RV) dysfunction (TAPSE < 15 mm) was found. LVAD logged several low-flow alarms (1.5 L/min, 8000 rpm), suggesting that a major device failure could be the trigger of this condition. Considering the unstable clinical setting and the past medical history of OGO [8], computed tomography (CT) was deferred and the patient was directly transferred to the catheterization laboratory to perform an urgent invasive diagnostic assessment. The procedure was performed under general anesthesia and with continuous transesophageal echocardiography monitoring. In order to avoid possible circulatory collapse during the procedure, both the arterial and venous femoral accesses were surgically obtained with the purpose of rapidly deploying veno-arterial extracorporeal membrane oxygenation (ECMO) assistance, if necessary. Via the right femoral artery, a first attempt at performing angiography using a diagnostic pig-tail catheter failed. Considered the presence of a known kinking site [8], an 8F 110 cm-long sheet (Flexor^®^Ansel Guiding Sheath, Cook Medical, Bloomington, IN, USA) was conducted through the distal anastomosis of the outflow graft to provide further support. Following this, a 130 cm-long supporting catheter (TrailBlazer™, Medtronic) over standard 0.035″ × 260 cm hydrophilic guidewire was advanced beyond the kinking and a diagnostic angiography was performed. As showed in Figure 2, a discrete filling defect located at the kinking site was found, suggesting recurrent thrombosis.

The first guidewire was removed and a 0.035″ × 260 cm stiff guidewire (Amplatz Super Stiff™, Boston Scientific, Natick, MA, USA) was deployed distally. Following this, an “anchoring technique” [8] with a 12 mm balloon located in the proximal tract of the outflow graft was adopted in order to advance the long sheet throughout the kinking. After that, a balloon-expandable 10 × 79 mm endoprosthesis (GORE^®^ Viabahn^®^ VBX) was implanted. Distal anchoring and stent positioning are shown in Figure 3A,B.

Due to severe underexpansion of the implanted stent, several postdilations were performed using peripheral balloons of increasing diameters (MUSTANGTM 0.035″, Boston Scientific) (see Figure 4).

Final angiography showed complete flow restoration and the optimal patency of the stent previously implanted in the proximal segment of the graft 12 months before, as shown in Figure 5.

Despite the percutaneous treatment being effective in improving LVAD parameters (4 L/min, 4500 rpm), hemodynamic instability persisted. Considering the severe RV failure recently found, a percutaneous RV support device (RVSD) was placed via right femoral vein and right internal jugular vein in order to stabilize the patient [7]. Notably, further improvement of LVAD parameters following the RVSD positioning was evident (6 L/min, 6000 rpm), confirming the key role played by the RV failure in this specific setting. During the following days in the ICU, LVAD parameters showed stable values and hemodynamics progressively improved. The postprocedural phase was complicated by severe pneumonia, which led to a further destabilization of the clinical profile. The blood culture revealed the presence of a carbapenem-resistant Klebsiella pneumoniae (Cr-KPN) as potential causative bacteria of the pneumonia. Antibacterial therapy with a combination of ceftazidime and avibactam was promptly started.

Nevertheless, the patient died due to refractory septic shock 10 days after the procedure.

## 3. Discussion

Following the results of major randomized trials [9], LVADs started playing a core role in the management of advanced HF. Nevertheless, conflicting results have been reported regarding the incidence of LVAD failure. In the “Evaluating the HeartMate 3TM with Full MagLev Technology in a Post-Market Approval Setting” (ELEVATE) registry and the “Multicenter Study of MagLev Technology in Patients Undergoing Mechanical Circulatory Support Therapy with HeartMate 3” (MOMENTUM 3) trial, reported LVAD thrombosis rates were, respectively, 0% and 1.4% [3,9]. However, evidence coming from recent observational studies suggests that its incidence could be higher [5,10]. Compared to other possible sites of thrombosis (such as pump thrombosis), OGO appears to be less frequent, rarely associated with signs of hemolysis and commonly anticipated by symptoms of worsening HF [7]. Different etiologies of OGO have been described, including isolated thrombosis, kinking, outflow torsion, external bend relief compression and others [11]. In these patients, percutaneous treatment can be highly effective in restoring LVAD function and avoiding high-risk surgical redo [10,12,13], even if highly complex anatomical scenarios can be encountered [8].

Despite CT is often fundamental in confirming the diagnosis [7], different cases of CT-deferring have been reported in the literature, especially when OGO is strongly suspected [5]. In our case, the severely impaired clinical conditions forced the physicians to choose a time-saving strategy and defer to preprocedural diagnostic assessment. Overall, we believe that an invasive diagnostic strategy should be preferred in those patients with suspected recurrent OGO if performed by trained and experienced operators, especially in an urgent setting.

As we previously reported [8], VA-ECMO assistance can play a core role during the percutaneous treatment of OGO. Indeed, during the procedure, the flow across the LVAD is often interrupted (in order to avoid procedural complications) and this can potentially lead to a critical reduction in the cardiac output. Moreover, kinkings, torsions and several others anatomical hurdles can lead to longer and more complex interventions, increasing the procedural risk. In case of recurrent OGO, previous anatomical and procedural scenarios should be considered by operators performing the procedure, possibly leading to a reduction in the procedural time and an increase in the chances of success. In our case, despite VA-ECMO assistance being strongly considered, we managed to perform the intervention using the same strategies previously adopted 12 months before (as the “distal balloon anchoring” technique) [8], effectively reducing the procedure-related risk and time (50′). Nevertheless, patients with LVAD and OGO are exposed to an extremely high risk of adverse events and, even if LVAD flow is effectively restored, other cardiac and noncardiac complications can occur. In our case, RV failure and pneumonia led to a negative clinical outcome despite the effectiveness of the percutaneous treatment. Interestingly, right ventricular decompensation and untreatable infections were, respectively, the first and the second causes of death at 6 months in the ELEVATE registry (45.7% and 20.2%, respectively) [3]. Similar results were described in the MOMENTUM 3 trial [9]. Future observational studies will address the relative impact of cardiac and noncardiac causes of death in defining the in- and out of- hospital outcomes of these patients.

Only a few studies have reported long-term clinical and technical results in patients with LVAD previously treated with a percutaneous approach due to OGO. In a recently published work, Agrawal and colleagues reported data from 20 patients (6.2%) out of 322 with LVAD suffering OGO and treated with a percutaneous strategy. According to their results, the procedure was safe and effective in most cases, with recurrent OGO occurring in only two patients over a median follow-up of 15 ± 10 months. Interestingly, in both cases OGO was located in a different previously nonstented site [10]. Similarly, in our reported case, recurrent OGO occurred in correspondence with a previously nonstented kinking site, confirming both the role of kinking in OGO etiology and the long-term effectiveness of the percutaneous treatment in terms of stent’s patency. Currently, some authors suggest that a strategy involving routine aggressive stenting of the entire outflow graft should be preferred in order to reduce the rates of recurrent OGO [10], but no definitive evidence is available.

Choosing the best pharmacological strategy can be extremely hard in this setting. In major studies investigating outcomes of patients with LVAD, significant bleeding events seem to be extremely common (comprised between 20% and 40%) [3,9]. Despite graft thromboses being rare in these studies, it is reasonable to suppose that the relative ischemic risk for patients who experience OGO could be higher (especially if recurrent). Following the first procedure, our patient was the first candidate to be treated with aspirin and vitamin K antagonist (VKA) in order to prevent stent thrombosis. However, during follow-up, the antiplatelet therapy was suspended due to a major neurological bleeding. It could be speculated that, in our case, VKA therapy was effective in holding the stent’s patency but failed in avoiding further thrombosis in other nonstented sites of the outflow graft. This could confirm the core role played by anatomical features, as severe kinking, in favoring thrombosis and OGO. The clinical relevance of this combination of antiplatelet and antithrombotic therapies should be confirmed in larger future studies.

## 4. Conclusions

Percutaneous intervention of OGO is feasible and effective. Different considerations could be advanced by our experience, as follows: (1) recurrent OGO, although uncommon, can be effectively treated with a percutaneous redo. In such cases, an invasive diagnostic algorithm with a CT deferral could be chosen, especially if OGO is highly suspected and clinical conditions are unstable; (2) isolated VKA therapy may be effective in preserving long-term stent’s patency, but can fail in preventing further thrombosis in nonstented segments, highlighting the role played by the anatomical features of the outflow graft (as kinkings); and (3) despite procedural success often being achieved, major comorbidities as RV failure and infective complications can largely impair a patients’ prognosis. Overall, patients suffering LVAD failure due to OGO must be considered at extremely high risk of adverse events. Nevertheless, percutaneous treatment could potentially represent a standard care option in the near future. Further studies are needed to address the impact of cardiac and noncardiac causes of death in patients with OGO and to coherently define the best combination of antiplatelet/antithrombotic strategies, in conjunction with the high ischemic and bleeding risk.

## Figures and Tables

**Figure 1 ijerph-19-05976-f001:**
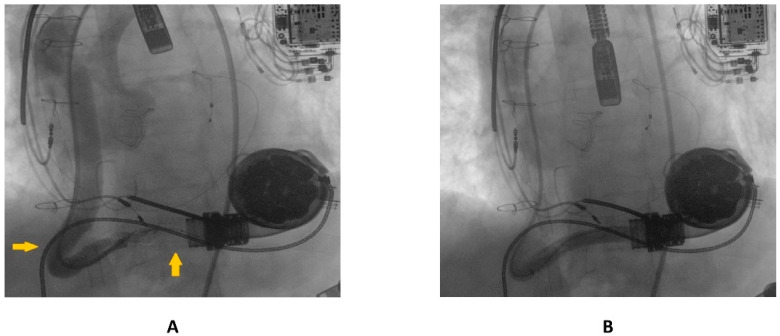
(**A**) preprocedural angiography of the first outflow graft obstruction episode (yellow arrows highlighting the kinking site and proximal thrombosis location); (**B**) result following the first percutaneous intervention.

**Figure 2 ijerph-19-05976-f002:**
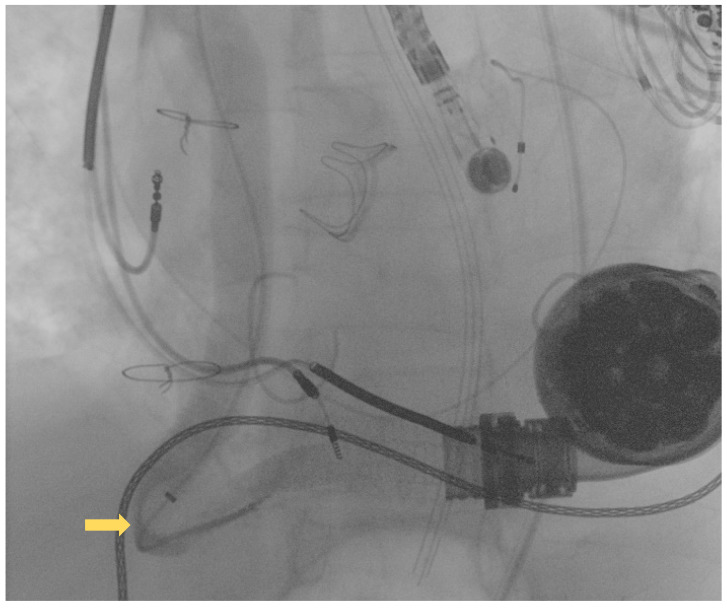
Diagnostic angiography showing recurrent outflow graft obstruction (yellow arrow).

**Figure 3 ijerph-19-05976-f003:**
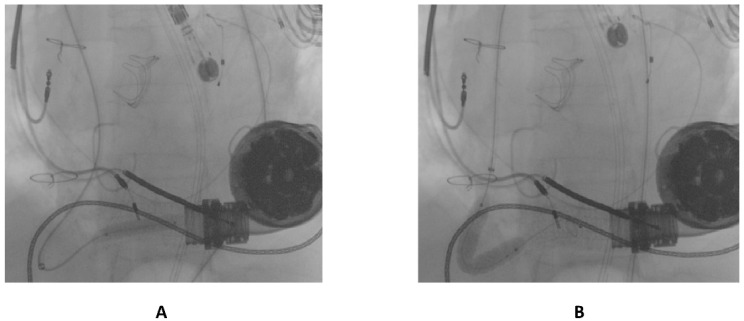
(**A**) distal anchoring; (**B**) stent deployment.

**Figure 4 ijerph-19-05976-f004:**
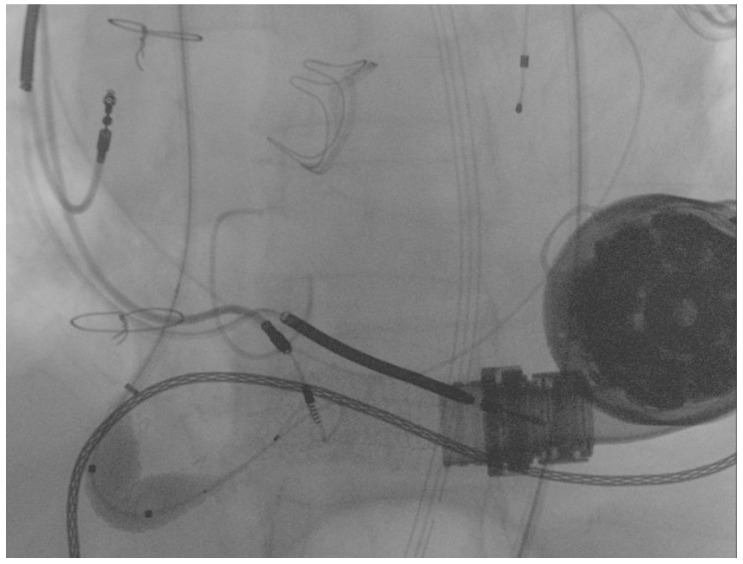
Postdilation of the implanted stent.

**Figure 5 ijerph-19-05976-f005:**
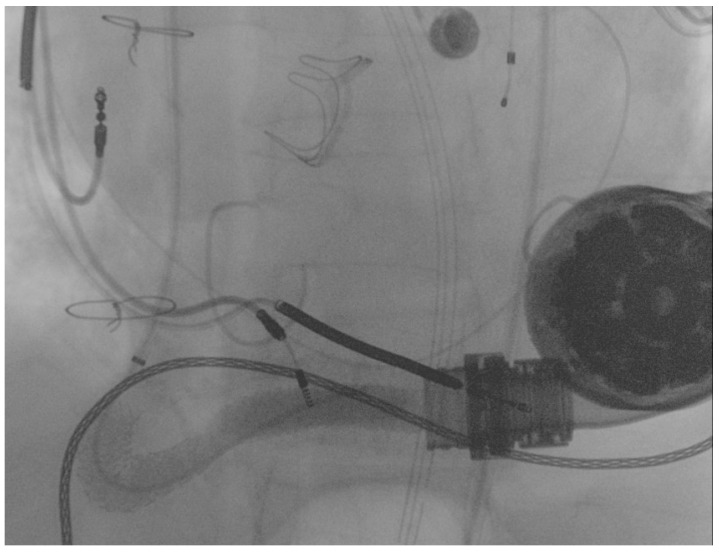
Final result following the percutaneous intervention.

## Data Availability

Data regarding this clinical case can be provided by the authors following reasonable request.

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
