# Peer review of "A Redo Percutaneous Emergency Intervention of Left Ventricular Assist Device Graft Occlusion"

_ijerph, 2022, doi:10.3390/ijerph19105976_

Round 1

Reviewer 1 Report

Line 28: ¨other non-cardiac conditions¨instead ¨several non-cardiac condition¨

Line 37: ¨impaired ejection fraction¨ instead ¨advanced HF¨

Line 40: ¨Despite being considered an uncommon event ...¨instead

 ¨Despite firstly considered an uncommon event¨

Line 49: ¨A 69 years old male with advanced HF...¨àit is important to describe the number of the LVEF before everything happened

Line 50: it is important to write when the symptoms started

Line 81: ¨A Flexor... stent was conducted through...¨instead ¨(Flexor® Ansel Guiding Sheath, Cook Medical) was advanced through the distal anastomosis...¨

Line 146: ¨should be considered¨instead ¨should be already known¨

Line 179: ¨However, during follow-up, the antiplatelet therapy was suspended due to a major neurological bleeding¨-->this should be written in the Case Report, not in the Discussion

Line 184: ¨The clinical relevance of this combination of antiplatelet and antithrombotic should be confirmed in larger future studies ¨instead ¨Larger observational studies are mostly needed to address¨

Author Response

We do thank the reviewer for his comments. Our corrections are listed below: 

Line 28: ¨other non-cardiac conditions¨instead ¨several non-cardiac condition¨

-Addressed

Line 37: ¨impaired ejection fraction¨ instead ¨advanced HF¨

-Addressed

Line 40: ¨Despite being considered an uncommon event ...¨instead

 ¨Despite firstly considered an uncommon event¨

-Addressed

Line 49: ¨A 69 years old male with advanced HF...¨it is important to describe the number of the LVEF before everything happened

-A dedicated statement has been added at row 49

Line 50: it is important to write when the symptoms started

-A dedicated statement has been added at rows 51-52

Line 81: ¨A Flexor... stent was conducted through...¨instead ¨(Flexor® Ansel Guiding Sheath, Cook Medical) was advanced through the distal anastomosis...¨

-Addressed

Line 146: ¨should be considered¨instead ¨should be already known¨

-Addressed

Line 179: ¨However, during follow-up, the antiplatelet therapy was suspended due to a major neurological bleeding¨-->this should be written in the Case Report, not in the Discussion

-A dedicated statement has been added at rows 64-66

Line 184: ¨The clinical relevance of this combination of antiplatelet and antithrombotic should be confirmed in larger future studies ¨instead ¨Larger observational studies are mostly needed to address

-Addressed

Reviewer 2 Report

The submitted manuscript is a clinical case report. 
The case report - was confirmed by patient consent for treatment and described in a scientific report. It describes an important aspect which is: A Redo Percutaneous Emergency Intervention Of Left Ventricular Assist Device Graft Occlusion.

In the Introduction, the authors discuss the general subject matter. The case report chapter contains all the important information about the patient, the qualification, the treatment and the stages after the intervention. The authors enriched this chapter with photographs, which increases the value of the manuscript. In the Discussion chapter the authors compare their report with the opinions and indications of other researchers. The discussion is extensive and constitutes a constructive and consistent whole of the manuscript. The authors conclude with their conclusions. In this case, I believe the conclusions should be more structured and condensed (preferably bulleted). Final conclusion: Future studies are needed to address the impact of cardiac and non-cardiac causes of death in patients with OGO and to define the best combination of antiplatelet/antithrombotic strategy, coherently with the high ischemic and bleeding risk.   - should be highlighted. Additionally, I recommend a separate section for clinical recommendations. 

Author Response

We do thank the reviewer for his comments

The "Conclusions" paragraph has been revised according to his suggestions.

Considered that our paper is a simple case report, no definitive conclusions should be drawn from it. Following, we do not believe that specific clinical recommendations should be provided to eventual readers. However, we agree that a "bulleted list" version of the "Conclusion" paragraph can better highlight the core messages of our isolated experience.

Reviewer 3 Report

The authors reported a case of emergent percutaneous intervention of recurrent left ventricular assist device graft occlusion. This paper is very interesting but has several concerns.

Major comments

  1. The authors’ percutaneous treatments including implantation of RVSD were successful. However, his prognosis was very poor because of septic shock 10 days after the treatment. The authors mentioned untreatable infection was the second cause of death on page6, line 155. What were the causative bacteria (a result of blood culture) of this case? How did you treat them? You had better mention the clinical course in detail. 
  2. What was the clinical implication of this case?

Minor comments

  1. On page 2, line 61, you had better add the words “yellow arrows” to the figure legend.
  2. You had better add the arrows to Figure 2.

Author Response

We do thank the reviewer for his comments.

-We provided details regarding the nature of the infective pneumonia/septic shock and our attemptives to treat them in the case report description. In particular we the blood culture revealed the presence of a Carbapenem-resistant Klebsiella Pneumoniae (Cr-KPN) as a potential causative bacteria of pneumonia. Antibacterial therapy with a combination of ceftazidime and avibactam has been promptly started.

-Regarding clinical implication, we do believe that no definitive recommendations should be derived from an isolated case report experience (see also our answer to reviewer 2). Despite we hope that our reported case will help operators facing this uncommon condition in future, only larger studies will be able to provide clear recommendations. 

On page 2, line 61, you had better add the words “yellow arrows” to the figure legend.

-Addressed

You had better add the arrows to Figure 2

-Addressed

Round 2

Reviewer 3 Report

Major comments:

1. This case has another problem of infection control. Because Carbapenem-resistant Klebsiella Pneumoniae is one of the causative bacteria of nosocomial infection.

Minor comments:

1. Though yellow arrow is mentioned in figure caption (line 93), it is not described in the Figure 2.

Author Response

Major comments:

  1. This case has another problem of infection control. Because Carbapenem-resistant Klebsiella Pneumoniae is one of the causative bacteria of nosocomial infection.

We do thank the Reviewer for his comment.

We believe that in this particular clinical case the readers will be attracted mostly by the interventional techniques that we adopted to overcome this rare condition. As a matter of facts, as interventional cardiologist, we focused our attention on the cardiological and interventional aspects of the case. After a confrontation with the anesthetist who follow the case with us, we decide to provide just the main information concerning the pneumonia as we did.

Minor comments:

1. Though yellow arrow is mentioned in figure caption (line 93), it is not described in the Figure 2: addressed